# Prevalence of hypertension and associated factors among adult residents in Arba Minch Health and Demographic Surveillance Site, Southern Ethiopia

**Adefris Chuka[1], Befikadu Tariku Gutema[2,3]\*, Gistane Ayele[2,3], Nega Degefa Megersa[4], Zeleke Aschalew Melketsedik[4], Tadiwos Hailu Zewdie[5]**

**1** CARE Ethiopia Hawassa Project Office, Hawassa, Ethiopia, **2** School of Public Health, Arba Minch University, Arba Minch, Ethiopia, **3** Arba Minch Health and Demographic Surveillance System (HDSS), Arba Minch, Ethiopia, **4** School of Nursing, Arba Minch University, Arba Minch, Ethiopia, **5** School of Medicine, Arba Minch University, Arba Minch, Ethiopia

\* befikadutariku2@gmail.com, befikadu.tariku@amu.edu.et

**Data Availability Statement:** All relevant data are within the paper and its Supporting Information files.

## Abstract

Hypertension is the leading risk factor for mortality and it is also one of the major risk factors for other non-communicable diseases (NCDs). The objective of the study was to assess the prevalence of hypertension and its associated factors among adults residing in Arba Minch health and demographic surveillance site (HDSS), Southern Ethiopia. A community-based cross-sectional survey was conducted in 2017 on the estimated sample size of 3,368 adults at Arba Minch Health and Demographic Surveillance site (HDSS). Data were collected using the WHO STEPS survey tools. Bivariate analysis was done to detect candidate variables at *P-value* less than 0.25 and entered into the final model to identify the independent predictors of hypertension. The prevalence of hypertension was 18.92% (95% CI: 17.63–20.28). The magnitude increase among respondents in the older age group [AOR 1.39 (95%CI: 1.05–1.84), 1.68 (95% CI: 1.26–2.23) and 2.67 (95%CI: 2.01–3.56) for age group 35–44, 45–54 and 55–64, respectively, compared to 25–34 years old group] and those with the higher wealth index [AOR 1.86 (95%CI: 1.33–2.59), 2.68 (95% CI: 1.91–3.75) and 2.97 (95%CI: 2.08–4.25) for 3rd quantile, 4th quantile and 5th quantile, respectively, compared to 1st quantile]. The odds of hypertension reduce among married participants (AOR 0.66, 95% CI: 0.51–0.85). Respondents with overweight (AOR 1.44, 95%CI: 1.02–2.02), khat chewing (AOR3.31, 95%CI: 1.94–5.64), low fruit and/or vegetable consumption (AOR 1.27, 95%CI: 1.05–1.53) and those who do not use coffee and tea (AOR 1.52, 95%CI: 1.03–2.24) had significantly higher likelihood of hypertension. Nearly one out of five participants have hypertension in this population. As hypertension is one of the silent killers, it is advisable to develop a system for enabling early detection and monitoring in the older age groups and overweight individuals.

**Funding:** This study was funded by Arba Minch Demographic Surveillance and Health Research Center, Arba Minch University. The funder did not have a role in the study design, data collection and analysis, decision to publish, or preparation of the manuscript. Rather the funder cover the cost related to the data collection and analysis.

**Competing interests:** The authors have declared that no competing interests exist.

## Introduction

There has been a dramatic transition from infectious diseases to non-communicable diseases (NCDs) burden in low- and middle-income countries (LMICs) [1,2]. NCDs are wrongly perceived as affecting only the rich, yet a significantly large proportion of death in LMICs are from NCDs [2–4]. In Ethiopia, NCDs were the leading cause of mortality in 2015 after adjusted for differences in the age distribution of the population [5]. Ethiopia adopted the World Health Organization (WHO) global action plan on NCDs (2013–2020), which includes increasing public awareness and reducing the incidence of behavioral risk factors [6,7].

Hypertension is a major risk factor of cardiovascular diseases, which are the leading NCDs. It contributes to morbidity and mortality due to coronary heart disease, stroke, heart failure, peripheral vascular disease, renal impairment, retinal hemorrhages, and papilledema [8,9]. Nearly half of stroke and ischemic heart disease worldwide are attributable to high blood pressure [10]. WHO reports indicate that high blood pressure is one of the leading risk factors for mortality in the world [11,12]. LMICs are increasingly facing the double burden of diseases. In addition to the high prevalence of communicable diseases in LMICs, there is an increase in the risk of NCDs including high blood pressure [11]. From 1990 to 2015, the associated annual death due to high blood pressure (systolic blood pressure >140mmHg) increased from 97.9 to 106.3 per 100,000. Based on this projection, 14% of the total global deaths were attributed due to high blood pressure. In addition, there were 143 million Disability-adjusted life years due to high blood pressure [13].

One of the targets of Sustainable Development Goals was to reduce the prevalence of raised blood pressure by 25% in 2030 [14]. If not tackled timely through appropriate interventions, the NCDs will continue claiming the lives of many adults in the productive age group. In 2014, the Ethiopian government developed a strategic action plan for the prevention and control of NCDs. It indicates that the use of standardized screening, ensuring the availability and affordability of essential medicines and technologies for the prevention, diagnosis and treatment of NCD including hypertension are part of the strategies for the prevention and control of NCDs [7]. A study conducted in Ethiopia and a report by WHO showed that raised blood pressure is more common compared to other biological and behavioral risk factors for NCDs [4]. The study aimed to determine the prevalence and factors associated with hypertension among adults residing in Arba Minch health and demographic surveillance site (HDSS), which was collected based on the WHO STEP wise approach to Surveillance (STEPS).

## Materials and methods

### Study setting and area

The study was conducted in Arba Minch HDSS, which is located in Arba Minch Zuria District, Southern Ethiopia. Arba Minch Town, administrative town of the district is located 505 Km south from the capital city, Addis Ababa. In 2017, the total population of the Arba Minch Zuria district was 195,858 (50.01% female) [15]. Arba Minch HDSS includes nine Kebeles (the lowest administrative unit of Ethiopia)of Arba Minch Zuria District. Eight of the nine HDSS Kebeles are rural and the remaining one is semi-urban. According to a 2016 Arba Minch HDSS report, the total population of the site was 74,029 adults age 25–64 years.

### Study design, period and population

A community-based cross-sectional study was conducted from April to June 2017. The source population was adult residents (25–64 years) of Arba Minch HDSS. According to the 2016 HDSS report, 24,800 (nearly half of them were women) were eligible and included as source

population. Pregnant mothers and women who have a history of recent delivery up to 8 weeks were excluded from the study.

## Sample size determination and procedures

According to the WHO STEPS guideline for sample size calculation, the study population was categorized into eight groups based on the four age and sex categories [9]. The prevalence of hypertension in Durame Town, southern Ethiopia was considered for the calculation of the sample size, which was 22.4% [16]. Using a single population proportion formula and design effect of 1.5, the estimated sample size for a group was 396. With the consideration of a 5% non-response rate and eight groups to have an adequate level of precision for each age-sex estimate, the final sample size was 3,368. The sampling frame was extracted from Arba Minch HDSS database using sex, date of birth, individual and household identifications as extraction variables. Based on sex and age category (25–34 years, 35–44 years, 45–54 years and 55–64 years), the sampling frame was stratified into eight groups, and a random sampling technique was implemented using STATA version 14 to select the study participants from each stratum. From each stratum, 425 study participants were selected. The HDSS identification of the selected individuals (name and individual HDSS id) with their respective household (household HDSS id) were used to identify the study participants and their location.

## Data collection instruments

Data collection instruments were adapted from WHO STEPS instruments. From three levels of the STEPS approach, only step one and two were applied for this study. The first step is the questionnaire-based, which was designed to obtain core data on socio-demographic information, tobacco and alcohol use, fruit & vegetable intake and physical activity. Physical activity was measured using the Global Physical Activity questionnaire. Blood pressure, waist and hip circumference, height and weight measurement are included from the second step and measured according to the WHO STEPS approach [9,17]. Blood pressure measurements were taken using an Omron T9P digital automatic blood pressure monitor. Bodyweight was measured using a SECA digital scale and height was measured using a stadiometer. Waist circumference and hip circumference were measured using flexible constant tension tape. In addition to WHO STEPS, variables for the frequency of meat consumption, coffee and tea consumption, Khat chewing, wealth index and mental stress were included in the questionnaire. Household wealth index questions were obtained from the Ethiopia Demographic and Health Survey, which was based on the household ownership of the productive asset and household characteristics [18]. Mental stress was examined using the self-reporting questionnaire (SRQ-20), which is used for easily acquired mental health symptoms of the participants [19].

## Data collection process

Training was given for three days on data collection material and measurement procedures for 20 data collectors and four supervisors. The training was given on the interview technique, the content of the questionnaire and measurement techniques. The WHO STEPS approach was used to collect the data. Interviews and measurements took place in the respondents' dwelling. Three blood pressure readings were taken on the left upper arm with the participant in a seated position following at least 5 to 10 minutes of rest. The participants took rest for three minutes between the readings. Bodyweight (to the nearest 0.5 kg) was taken with the participant on bare feet and with light clothing using SECA digital scale (model number 877). Height (to the nearest 1 cm) was measured using a stadiometer with participants wearing no shoes and without headwear. Waist circumference measured at the midpoint between the palpable rib and

the iliac crest. For measuring the hip circumference, the greatest posterior protuberance of the buttocks with a constant tension tape was used while the subject stands with arms at the sides, feet positioned close together, and weight evenly distributed across the feet [9].

## Data quality control

Experienced data collectors and supervisors of Arba Minch HDSS have collected the data. The training was given for three days on data collection material and measurement procedures. The pre-test was conducted on 2% of the sample size and the result was used to modify the questionnaire. Supervisors checked the completeness and consistency of the collected data daily. Standardized measuring instruments were used for physical measurements. To increase the response rate, the data collectors were repeatedly visited (at least three times) those participants who were not present at the house during data collection time.

## Variables

The independent variables include socio-demographic status, anthropometric measurements, physical activity level, intake of meat, vegetable, fruit, coffee, tea and alcohol, and tobacco use. Socio-demographic variables included sex, age, residency, religion, ethnicity, marital status, educational level and wealth status. Wealth status was generated by computing wealth index using a principal component factor analysis based on the household ownership of the productive asset and household characteristics. The computed value then grouped in to five quantiles [18]. The anthropometric index includes body mass index (BMI) and Waist-to-hip ratio (WHR). BMI was generated by computing weight in kilograms per height in meters squared $(kg/m^2)$ and categorized in to underweight $(<18.5kg/m^2)$, normal $(18.5–24.9kg/m^2)$ and overweight $(\geq25kg/m^2)$. Waist-to-hip ratio (WHR) was calculated by dividing waist circumference (cm) to hip circumference (cm) and grouped in to normal for those with WHR below 1.00 for men and 0.85 for women [9]. Physical activity levels with average metabolic equivalents-minutes per week < 600, 600–3000 and >3000 were grouped in to low, moderate and high, respectively [9,17]. Based on the self-reporting questionnaire (SRQ-20), mental stress was categorized in to three (mild, moderate, and severe) [19]. Meat consumption was assessed by asking the frequency of the different types of meat consumptions and grouped staring from day to year. The consumption of fruits and vegetables were assessed by asking the serving size of the consumption per day within a week. Khat chewing, and coffee and tea consumption were assessed by asking their use. Heavy alcohol consumption was defined as the consumption of 6 or more drinks on a single occasion at least a month [20]. The current tobacco consumption was defined as the current use of smoked and/or smokeless tobacco. The mean of all three blood pressure readings was used to determine systolic and diastolic blood pressures. The independent variable, hypertension, was defined as an average systolic blood pressure of 140 mmHg or higher and/or average diastolic blood pressure of 90mmHg or higher and/or a participant taking anti-hypertension medication within the preceding two weeks [9,21].

## Ethical statement

Ethical approval was obtained from the Institutional Review Board of Arba Minch University. A formal letter was submitted to the concerned bodies to get permission to conduct the research in the settings. Verbal informed consent was obtained from study participants before the interview and physical measurements. The privacy of the study participants was maintained by interviewing the participants alone. Those identified as having high blood pressure were referred to the nearby health facilities for further diagnosis and treatment.

## Data processing and analysis

EPI-data version 3.1 statistical software was used for data entry and the data were exported to STATA version 14 for data management and analysis. To assess the presence of an association between dependent and independent variables, bivariate analysis was done and variables with p-value less than 0.25 were entered into a multiple logistic regression model to identify the independent predictors of hypertension. To assess the presence of an association between dependent and independent variables, a p-value of less than 0.05 was considered. For the assessment of multicollinearity, variable inflation factors were used and the maximum value was 1.45. Model fitness was checked using Hosmer and Leme show goodness of fit tests and it was 5.95 (P = 0.653).

## Results

### Socio-demographic characteristics of the study population

Totally 3,346 adults were enrolled in the study, with a response rate of 99.35%. Half of the participants were female (49.97%). The mean (SD) age of the participants was 44.59 (11.17) years with 44.80 (11.07) and 44.38 (11.27) years for men and women, respectively. Most of the study participants were married (87.90%), from Gamo ethnic group (81.08%), and no formal education (69.75%) (Table 1).

Table 1. Socio-demographic characteristics of the study participants by hypertension.

| Variables | Categories | Hypertension | | Total |
|---|---|---|---|---|
| | | No | Yes | |
| Sex | Male | 1375(82.14) | 299(17.86) | 1674(50.03) |
| | Female | 1338(80.02) | 334(19.98) | 1672(49.97) |
| Age group | 25–34 | 678(84.33) | 126(15.67) | 804(24.03) |
| | 35–44 | 721(83.55) | 142(16.45) | 863(25.79) |
| | 45–54 | 688(81.42) | 157(18.58) | 845(25.25) |
| | 55–64 | 626(75.06) | 208(24.94) | 834(24.93) |
| Residency | Rural | 2,311(82.51) | 490(17.49) | 2,801(83.71) |
| | Urban | 402(73.76) | 143(26.24) | 545(16.29) |
| Ethnicity | Gamo | 2214(81.61) | 499(18.39) | 2713(81.08) |
| | Zeyse | 236(81.38) | 54(18.62) | 290(8.67) |
| | Other | 263(76.68) | 80(23.32) | 343(10.25) |
| Religion | Protestant | 1704(80.99) | 400(19.01) | 2104(62.88) |
| | Orthodox | 860(81.59) | 194(18.41) | 1054(31.50) |
| | Other | 149(79.26) | 39(20.74) | 188(5.62) |
| Marital status | Unmarried | 281(69.38) | 124(30.62) | 405(12.10) |
| | Married | 2432(82.69) | 509(17.31) | 2941(87.90) |
| Educational status | No formal | 1920(82.26) | 414(17.74) | 2334(69.75) |
| | Primary school | 608(79.17) | 160(20.83) | 768(22.95) |
| | Secondary & above | 185(75.82) | 59(24.18) | 244(7.29) |
| Wealth Index | $1^{st}$ quantile | 599(89.40) | 71(10.60) | 670(20.02) |
| | $2^{nd}$ quantile | 579(86.29) | 92(13.71) | 671(20.05) |
| | $3^{rd}$ quantile | 548(82.16) | 119(17.84) | 667(19.93) |
| | $4^{th}$ quantile | 501(74.89) | 168(25.11) | 669(19.99) |
| | $5^{th}$ quantile | 486(72.65) | 183(27.35) | 669(19.99) |

**Table 2. Physical measurements and behavioral characteristics of the study participants by hypertension.**

| Variables | Categories | Hypertension | | Total |
|---|---|---|---|---|
| | | **No** | **Yes** | |
| BMI | Underweight | 324(77.33) | 95(22.67) | 419(12.52) |
| | Normal | 2141(83.44) | 425(16.56) | 2566(76.69) |
| | Overweight | 248(68.70) | 113(31.30) | 361(10.79) |
| WHR | Normal | 1611(82.49) | 342(17.51) | 1953(58.37) |
| | High | 1102(79.11) | 291(20.89) | 1393(41.63) |
| Physical activity level | Low | 621(78.51) | 170(21.49) | 791(23.64) |
| | Moderate | 326(80.49) | 79(19.51) | 405(12.10) |
| | High | 1766(82.14) | 384(17.86) | 2150(64.26) |
| Mental stress level | Mild | 1812(81.66) | 407(18.34) | 2219(66.32) |
| | Moderate | 801(80.50) | 194(19.50) | 995(29.74) |
| | Sever | 100(75.76) | 32(24.24) | 132(3.95) |
| Fruit & vegetable consumption | More than 5 serving | 1732(83.51) | 342(16.49) | 2074(61.98) |
| | Less than 5 serving | 981(77.12) | 291(22.88) | 1272(38.02) |
| Frequency of meat consumption | 1–4 times per week | 100(78.13) | 28(21.88) | 128(3.83) |
| | 1–3 times per month | 357(78.46) | 98(21.54) | 455(13.60) |
| | 1–4 time per year | 2014(82.74) | 420(17.26) | 2434(72.74) |
| | Never | 242(73.56) | 87(26.44) | 329(9.83) |
| Coffee and tea consumption | Coffee only | 232(78.64) | 63(21.36) | 295(8.82) |
| | Coffee & leaf tea | 2071(82.02) | 454(17.98) | 2525(75.46) |
| | Coffee & coffee leaf | 113(83.09) | 23(16.91) | 136(4.06) |
| | None | 297(76.15) | 93(23.85) | 390(11.66) |
| Current Tobacco consumption | No | 2167(81.19) | 502(18.81) | 2669(79.77) |
| | Yes | 546(80.65) | 131(19.35) | 677(20.23) |
| Heavy alcohol consumption | No | 2402(81.09) | 560(18.91) | 2962(88.52) |
| | Yes | 311(80.99) | 73(19.01) | 384(11.48) |
| Khat chewing | Yes | 35(53.03) | 31(46.97) | 66(1.97) |
| | No | 2678(81.65) | 602(18.35) | 3280(98.03) |

BMI: Body mass index; WHR: Waist-to-hip ratio.

More than one out of ten were underweight (12.52%) and overweight (10.79%). A large proportion of the study participants had central obesity (41.63%) based on WHR. Nearly 2% (66) of the study participants chew khat and one out of five (20.23%) use tobacco. Nearly 10% (329) of the study participants never consumed meat for a year (Table 2).

## Prevalence of hypertension

The prevalence of high blood pressure among study participants based on systolic blood pressure above 140mmHg and diastolic blood pressure above 90mmHg were 14.05% (95%CI: 12.91–15.27) and 12.43% (95%CI: 11.35–13.59), respectively. In general, those who had increased blood pressure (above 140mmHg systolic and/or 90mmHg diastolic blood pressure) was 16.86% (95%CI: 15.62–18.16). In addition, 4.03% (135) reported using anti-hypertensive medications during the data collection period. Out of those who use anti-hypertensive medications, 51.11% (69) had normal blood pressure on measurement. The overall prevalence of hypertension among participants was 18.92% (95%CI: 17.63–20.28). Among male and female participants, the prevalence of hypertension was 17.86% (95%CI: 16.10–19.77) and 19.98% (95%CI: 18.13–21.96), respectively.

The mean (SD) systolic and diastolic blood pressure were 122.39 (19.29) and 76.27(11.40) mmHg, respectively. The mean (SD) systolic blood pressure for men and women were 122.33 (17.72) and 122.44 (20.75) mmHg, respectively. For diastolic blood pressure, the mean (SD) for men and women were 76.30(11.78) and 76.24(11.01) mmHg, respectively. There was no significant difference between mean systolic ($P$ = 0.859) and diastolic ($P$ = 0.877) blood pressure among men and women.

## Factors associated with hypertension

Based on multiple logistic regression analysis, the likelihood of hypertension increased among the older age groups, unmarried, overweight, khat chewing, those with low intake of fruit and/ or vegetable, those who did not take coffee, coffee leaf and/or tea, and from higher wealth index households. The odds of being hypertensive among the age group of 35–45, 45–54 and 55–64 years were 1.39 (95%CI: 1.05–1.84), 1.68 (95%CI: 1.26–2.23) and 2.67 (95%CI: 2.01–3.56), respectively and these age groups were significantly higher compared to 25–34 years. Married participants were less likely (AOR 0.66, 95%CI: 0.51–0.85) to develop hypertension compared to unmarried. Concerning wealth index of the participants' household, as the wealth status goes to a higher level [3rd quintile (AOR 1.86, 95%CI: 1.33–2.59), 4th quintile (AOR 2.68, 95%CI: 1.91–3.75), and 5th quintile (AOR 2.97, 95%CI: 2.08–4.25)], the odds of developing hypertension increased compared to the 1st quintile. The likelihood of hypertension was higher among overweight (AOR 1.44, 95%CI: 1.02–2.02) compared to underweight participants. The odds of being hypertensive among participants who chew khat was 3.31 times (95%CI: 1.95–5.62) more likely as compared to those who did not. Consumption of low fruit and/or vegetable were 1.27 (95%CI: 1.05–1.53) times more likely to develop hypertension. The odds of being hypertensive was increased by 52% among participants who did not consume coffee, coffee leaf and/or tea (AOR 1.52, 95%CI: 1.03–2.24) as compared to those who consume coffee only (Table 3).

## Discussion

The prevalence of hypertension among the study participants was 18.92%. Community-based studies conducted among adult Ethiopians showed that the prevalence of hypertension was from 16.0–28.3% [16,22–27]. Most of the studies conducted in the Ethiopian adult population had a similar prevalence of hypertension with this finding [16,23,24,26]. However, studies conducted predominantly on urban population and study subjects age 35 years and more have reported a higher prevalence of hypertension. For instance, the prevalence of hypertension among the population at Dire Dawa City, Eastern Ethiopia (24.43%), Bedele Town, Southwest Ethiopia (24.8%) and Gondar, Northwest Ethiopia (28.3%) showed a higher prevalence than this study [25,27,28]. The finding of a meta-analysis by Kibret & Mesfin showed that the prevalence of hypertension in Ethiopia was 19.6% [29], which is similar to this study's finding. But the WHO global status report on NCDs in 2014 indicated that the raised blood pressure among Ethiopians were 24.0% [12].

In this study, the odds of hypertension increased among the older age groups. This finding is consistent with most of the studies conducted in Ethiopia and elsewhere [16,22,25–27]. Aging and hypertension are related to changes in arterial and arteriolar stiffness, which lead to the development of hypertension [30,31].

This report found that the odds of hypertension reduced among married adults. Similar to this finding, a study conducted among adult Iranians showed that the risk of hypertension increases among those who never married [32]. Previous studies indicated possible explanations for the effect of marital status on hypertension. Adherence to physical activity, diet, better

**Table 3. Multiple logistic regression analysis of determinants of hypertension among study participants.**

| Variables | Categories | COR | AOR | 95%CI |
|---|---|---|---|---|
| Age group | 25–34 | | Reference | |
| | 35–44 | 1.06 | 1.39* | 1.05–1.84 |
| | 45–54 | 1.23 | 1.68** | 1.26–2.23 |
| | 55–64 | 1.79** | 2.67** | 2.01–3.56 |
| Residency | Rural | | Reference | |
| | Urban | 1.68**** | 1.11 | 0.87–1.43 |
| Marital status | Unmarried | | Reference | |
| | Married | 0.47** | 0.66** | 0.51–0.85 |
| Educational status | No formal | | Reference | |
| | Primary school | 1.22 | 0.92 | 0.72–1.17 |
| | Secondary & above | 1.48* | 1.00 | 0.69–1.46 |
| Wealth Index | The poorest | | Reference | |
| | 2$^{nd}$ quantile | 1.34 | 1.38 | 0.98–1.93 |
| | Middle quantile | 1.83** | 1.86** | 1.33–2.59 |
| | 4$^{th}$ quantile | 2.83** | 2.68** | 1.91–3.75 |
| | The richest | 3.18** | 2.97** | 2.08–4.25 |
| BMI | Underweight | | Reference | |
| | Normal | 0.68** | 0.80 | 0.61–1.04 |
| | Overweight | 1.55** | 1.44* | 1.02–2.02 |
| Physical activity level | Low | | Reference | |
| | Moderate | 0.89 | 0.88 | 0.64–1.20 |
| | High | 0.79* | 0.90 | 0.73–1.12 |
| Mental stress level | Mild | | Reference | |
| | Moderate | 1.08 | 1.11 | 0.91–1.36 |
| | Sever | 1.42 | 1.30 | 0.84–1.99 |
| Fruit & vegetable consumption | More than 5 serving | | Reference | |
| | Less than 5 serving | 1.5** | 1.27* | 1.05–1.53 |
| Frequency of meat consumption | 1–4 times per week | | Reference | |
| | 1–3 times per month | 0.98 | 0.86 | 0.52–1.43 |
| | 1–4 time per year | 0.74 | 0.83 | 0.52–1.31 |
| | Never | 1.28 | 1.11 | 0.66–1.85 |
| Coffee and tea consumption | Coffee only | | Reference | |
| | Coffee & leaf tea | 0.81 | 1.02 | 0.74–1.40 |
| | Coffee & coffee leaf | 0.75 | 0.74 | 0.43–1.29 |
| | None | 1.15 | 1.52* | 1.03–2.24 |
| Khat chewing | No | | Reference | |
| | Yes | 3.94** | 3.31** | 1.95–5.62 |

\* p-value <0.05

\*\* p-value <0.01; AOR: adjusted odds ratio; BMI: Body mass index; CI: confidence interval; COR: crude odds ratio.

sleep, moods, lower self-rated symptoms, less stress and low night-time systolic blood pressure were more common among married couples. That might have an effect on reduced odds of hypertension among married couples [33,34]. Contrarily, a population and health survey among Ghana women showed that the likelihood of hypertension decreased among those who never married [35].

This report showed that the likelihood of developing hypertension increases among individuals from a wealthy household. A study conducted among low-income Mexican women indicated that systolic blood pressure significantly increases among those from higher-income, good housing, and asset index households [36]. Similarly, studies conducted in Kenya, Sudan and Ghana showed a significantly higher risk of hypertension among higher asset index households [37–39]. This relationship was also identified in studies conducted in middle-income countries like Serbia and India [40,41]. In developing countries, there is a nutritional transition that incorporates changes in the composition of diets richer in calories, salt, sugar, and saturated fats largely from animal sources, which showed the change in habitual diet [42,43]. In countries like Ethiopia, the cost of food richer in calories especially animal diet is higher and easily accessible for richer families mainly [44]. That may contribute to the higher odds of hypertension among rich individuals. Contrary to this finding, a longitudinal cohort study among African Americans showed that the odds of hypertension lower among rich individuals [45]. A meta-analysis on socioeconomic status and hypertension indicate similar finding, which was an increase in the odds of hypertension among low socio-economic status. On further analysis, the relationship is consistent with that of some high-income countries (countries from USA/Canada and Europe) but inverse for some Africa countries [46]. In general, studies from LMICs showed the likelihood of hypertension increase among individuals with high socio-economic status whereas, for those from high-income countries, the relationship is the inverse.

The likelihood of hypertension is higher among overweight adults in this study. Different studies support the relationship between BMI and increased blood pressure and/or hypertension [22,25,26,47,48]. Further, a report by Doll *et al* from different countries showed that BMI is associated with increased blood pressure independent of age. Such finding is similar across developed and developing countries. The finding by Doll *et al* also further indicated the association between increased blood pressure and BMI is independent of body fat distribution [49].

Diet plays a significant role in the prevention of the incidence of hypertension. Increased fruit and vegetable consumption are the major components of the dietary factor for the prevention of hypertension [16,50–53]. This study also revealed that the increased consumption of fruit and vegetable reduced the likelihood of hypertension among the study participants. Similarly, meta-analysis reports showed that increasing the consumption of fruit and vegetables prevented the risk of developing hypertension [54,55].

Khat (*Catha edulis)* is a chewable fresh-leave used as stimulant commonly in East Africa. It contains amphetamine-like stimulatory, which is a closely similar effect with cathinone [56,57]. In this report, the odds of hypertension increase among those who chew Khat. Studies conducted on the effect of Khat on hypertension also showed a similar association with this finding [58–60].

This report indicated that those who did not drink coffee or tea were at increased risk of developing hypertension. Different studies reported different findings of coffee and hypertension. Some of the studies concluded that moderate consumption of coffee is associated with an increase in the incidence of hypertension [61]. Other studies also showed a relationship between coffee intake and high blood pressure [62–64]. A report by Uiterwaal *et al.* indicated that there are no associations between the change of coffee intake and blood pressure [65]. Other findings indicated that low (less than one cup/week) or high (more than two cups/day) intake of coffee, not moderate, is associated with reduced risk of hypertension [61]. Some of the above-mentioned conclusions support a common recommendation by health care professionals to reduce or stop the consumption of coffee.

Some of the behaviors might vary seasonally (e.g., dietary intake) which may not be representative for whole year. In addition, the investigation does not capture the effect of some

behavioral modification after the development of hypertension such as reduction in coffee consumption.

## Conclusions

Nearly one out of five-adults developed hypertension in this population. This shows that the rural residents of Ethiopian population, like that of emerging economies elsewhere, are transitioning from infectious illness to chronic disease burden. Advance age, overweight, unmarried, khat chewing, low fruit and/or vegetable consumption, not taking coffee or tea and being from rich households are increasing the odds of having hypertension. As hypertension is one of the silent killers, it is advisable to develop a system for enabling early detection and monitoring in those with advanced age and overweight individuals. In addition, preventive measures of hypertension need to incorporate stopping khat chewing and reducing overweight and promoting fruit and/or vegetable consumption.

## Supporting information

**S1 Data.**
(CSV)

## Acknowledgments

We would like to extend our gratitude to the study respondents, field and data entry staff of the Arba Minch HDSS for their respective contributions to the production of the data used in this study. We acknowledge Arba Minch Demographic Surveillance and Health Research Center, Arba Minch University for providing sampling frame and supporting data collection, entry and analysis.

## Author Contributions

**Conceptualization:** Adefris Chuka, Befikadu Tariku Gutema, Gistane Ayele, Nega Degefa Megersa.

**Formal analysis:** Adefris Chuka, Befikadu Tariku Gutema, Nega Degefa Megersa.

**Funding acquisition:** Adefris Chuka, Befikadu Tariku Gutema.

**Investigation:** Adefris Chuka, Befikadu Tariku Gutema.

**Methodology:** Adefris Chuka, Befikadu Tariku Gutema, Gistane Ayele, Nega Degefa Megersa.

**Project administration:** Adefris Chuka, Befikadu Tariku Gutema.

**Resources:** Adefris Chuka, Gistane Ayele.

**Supervision:** Adefris Chuka, Befikadu Tariku Gutema.

**Writing – original draft:** Befikadu Tariku Gutema, Nega Degefa Megersa, Zeleke Aschalew Melketsedik, Tadiwos Hailu Zewdie.

**Writing – review & editing:** Adefris Chuka, Befikadu Tariku Gutema, Gistane Ayele, Nega Degefa Megersa, Zeleke Aschalew Melketsedik, Tadiwos Hailu Zewdie.

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
