## [Decision Letter · Decision Letter 0]

19 May 2020

PONE-D-20-02687

Prevalence of Hypertension and Associated Factors among Adult Residents in Arba Minch Health and Demographic Surveillance Site.

PLOS ONE

Dear Mr. Gutema,

Thank you for submitting your manuscript to PLOS ONE. After careful consideration, we feel that it has merit but does not fully meet PLOS ONE’s publication criteria as it currently stands. Therefore, we invite you to submit a revised version of the manuscript that addresses the points raised during the review process.

We would appreciate receiving your revised manuscript by Jul 02 2020 11:59PM. To enhance the reproducibility of your results, we recommend that if applicable you deposit your laboratory protocols in protocols.io, where a protocol can be assigned its own identifier (DOI) such that it can be cited independently in the future. For instructions see: http://journals.plos.org/plosone/s/submission-guidelines#loc-laboratory-protocols

We look forward to receiving your revised manuscript.

Kind regards,

Jianhong Zhou

Associate Editor

PLOS ONE

Journal Requirements:

3. Your ethics statement must appear in the Methods section of your manuscript. If your ethics statement is written in any section besides the Methods, please move it to the Methods section and delete it from any other section. Please also ensure that your ethics statement is included in your manuscript, as the ethics section of your online submission will not be published alongside your manuscript.

4. We note you have included a table to which you do not refer in the text of your manuscript. Please ensure that you refer to Table 3 in your text; if accepted, production will need this reference to link the reader to the Table.

Reviewers' comments:

Reviewer's Responses to Questions

**Comments to the Author**

1. Is the manuscript technically sound, and do the data support the conclusions?

Reviewer #1: Yes

Reviewer #2: Yes

2. Has the statistical analysis been performed appropriately and rigorously? 

Reviewer #1: Yes

Reviewer #2: No

3. Have the authors made all data underlying the findings in their manuscript fully available?

Reviewer #1: Yes

Reviewer #2: No

4. Is the manuscript presented in an intelligible fashion and written in standard English?

Reviewer #1: No

Reviewer #2: No

5. Review Comments to the Author

Reviewer #1: In the present manuscript the authors analyse the prevalence of arterial hypertension among adults residing in HDSS (Ethiopia). The prevalence was investigated in 3368 adults. The study has several limitations that must be addressed.

1. I suggest to use therm arterial hypertension instead of hypertension.

2. I think Country (Ethiopia) may be mentioned in the tittle of manuscript and abstract.

3. Perhaps hypertension is the leading risk factor for mortality, not leading risk...

4. In line 147: 44.8 (1107). Perhaps it should be 44.8 (11.07). Please check.

5. There are no p value in table 1 and 2.

6. There are no abbreviations under table 2 and 3.

7. In line 174: it’s enough to write p=0.859.

Reviewer #2: This paper addresses a major public health concern across the globe and will make an important contribution to the literature in this research area. However, before this paper can be deemed publishable, authors would have to address the following key concerns provided by sections of the paper.

Title:

Okay

Abstract

This section is quite okay. However, authors should avoid reporting odds ratios as prevalence as stated in the last sentence of the Results sub-section. Also, authors should provide the AORs for the older age and the higher wealth index as they did for the other variables.

Introduction

While this section is quite okay, there are some few issues authors need to address.

The sentence in lines 42 and 43 needs to be revised as this is not clear; particularly “the leading cause of premature age-standardized mortality rates”.

The 2001 statistics provided by the authors in lines 53-55 appear outdated. Authors should provide more recent figures, say 2015, as they did for the subsequent sentence in lines 55-56.

The sentence on line 60 needs to be revised for clarity and correctness.

Materials and Methods

In lines 66-71, while the authors did well by describing the geography of the study setting, it will be beneficial for them to also provide a few demographic descriptions, particularly the estimated population size of the study area.

In lines 86-88, it is not clear how the authors used Stata to randomly select participants from each stratum. Authors should be able to describe this process was done.

Lines 104-105. This sentence should be revised into a better form.

Authors did not describe all their variables of interest. It is more appropriate for authors to provide a separate sub-section to describe their outcome and predictor variables and their measurements.

For convenience, authors should consider labelling BMI as underweight (<18.0kg/m2), normal (18.0–24.9kg/m2), and overweight (≥25kg/m2) instead of the real measurements.

Authors stated that they used a p-value of 0.25 for the bivariate analysis but used 0.05 for the multivariate analysis. We know 0.25 is 75% while 0.055 is 95% CI. Authors should explain why they used these two different standards.

Authors have stated that they have assessed multicollinearity using variable inflation factors. Authors should provide the resultant VIF value like they did for the Hosmer and Lemeshow goodness of fit test.

Results

The presentation is generally okay but there are a few issues that need to be addressed.

In line 152, authors should rather say “more than one out of ten” instead of saying “nearly one out of ten”.

In line 155, authors seem to want to say, “fresh food” instead of “flesh food”. Also, there appear to be no fresh food in Table 2 as stated by the authors.

Authors should avoid using prevalence in the multivariate analysis section as can be found in line 155.

In line 182, authors should use “age groups” instead of “groups of age”.

In line 184, authors should delete the “to” in the “concerning to”.

Lines 187, 236, 277: Authors should avoid using “probability” and “prevalence” as they worked with only odds ratios.

For easy visibility and convenience, authors should use asterisks (*) to indicate the significant values in Table 3.

Discussion

In line 203 and 204, revise the sentence. Use “have reported” instead of “were reported”.

Authors should revise the sentence in lines 207 and 208 as there is something wrong with it.

Line 209: Ethiopians instead of Ethiopian.

Lines 210, 214, 230, 231, 233, 239, 254: Avoid prevalence here. Odds, chances, and likelihood are examples author may use.

Line 215: Iranians instead of Iranian.

Line 216, 217-218: Those who never married, not those never married.

Line 218 and others; Authors should explain the possible (speculative) reasons for their findings. For example, why would married women have low hypertension as found in your study? This should be done for the significant variables discussed in this section. Where this is not obvious, authors could briefly explain what could have led to this finding?

Line: 264: Reduced risk instead of reduced the risk.

Lines 266-267: Authors should either state and explain these factors or delete this statement as it appears incomplete and needless.

Lines 268-271: Authors should revise these sentences very well as they appear to be incomplete and unclear.

Conclusion

Lines 273-275: Authors should adequately revise this sentence as it is unclear.

General comments

While the paper has addressed an important public health issue and will contribute to the literature, it is noteworthy that the paper is also laden with some grammatical and diction challenges that have constrained the quality and flow of the paper. These are mainly found in the introduction and the discussion sections. Hence, this paper should be adequately proofread and revised accordingly before it be publishable.

6. PLOS authors have the option to publish the peer review history of their article (what does this mean?). If published, this will include your full peer review and any attached files.

Reviewer #1: No

Reviewer #2: No

---

## [Author Response · Author response to Decision Letter 0]

5 Jun 2020

Summary 

The reviewer gave important comments. We modify based on the comment given. In addition, we attached document with trace changes and the revised. 

Response:- Corrected according to the guideline 

Response:- We are decided to upload the dataset with the document 

3. Your ethics statement must appear in the Methods section of your manuscript. If your ethics statement is written in any section besides the Methods, please move it to the Methods section and delete it from any other section. Please also ensure that your ethics statement is included in your manuscript, as the ethics section of your online submission will not be published alongside your manuscript.

Response:- Correction made according to the suggestion 

4. We note you have included a table to which you do not refer in the text of your manuscript. Please ensure that you refer to Table 3 in your text; if accepted, production will need this reference to link the reader to the Table.

Response:- Correction made according to the suggestion 

Review Comments to the Author

Reviewer #1: 

In the present manuscript the authors analyse the prevalence of arterial hypertension among adults residing in HDSS (Ethiopia). The prevalence was investigated in 3368 adults. The study has several limitations that must be addressed.

1. I suggest to use therm arterial hypertension instead of hypertension.

Response:- It is correct that we measured the arterial hypertension. But commonly description about hypertension related issues is about the arterial hypertension unless otherwise specified like pulmonary, portal etc. That is why we decided to use ‘hypertension’. 

2. I think Country (Ethiopia) may be mentioned in the tittle of manuscript and abstract.

Response:- We noted that and made the correction 

3. Perhaps hypertension is the leading risk factor for mortality, not leading risk...

Response:- Noted and made the correction 

4. In line 147: 44.8 (1107). Perhaps it should be 44.8 (11.07). Please check.

Response:- Noted and made the correction 

5. There are no p value in table 1 and 2.

Response:- On table 1 and 2, we only reported the descriptive information. That is why we do not include the p-value. Both the bivariate and multiple logistic regressions analysis is reported in table 3.

6. There are no abbreviations under table 2 and 3.

Response:- Noted and made the correction 

7. In line 174: it’s enough to write p=0.859.

Response:- Noted and made the correction 

Reviewer #2: 

This paper addresses a major public health concern across the globe and will make an important contribution to the literature in this research area. However, before this paper can be deemed publishable, authors would have to address the following key concerns provided by sections of the paper.

Title:

Okay

Abstract

This section is quite okay. However, authors should avoid reporting odds ratios as prevalence as stated in the last sentence of the Results sub-section. Also, authors should provide the AORs for the older age and the higher wealth index as they did for the other variables.

Response:- Based on the comment, we corrected the statements. 

Introduction

While this section is quite okay, there are some few issues authors need to address.

The sentence in lines 42 and 43 needs to be revised as this is not clear; particularly “the leading cause of premature age-standardized mortality rates”.

Response:- Yes there was error and we corrected it. 

The 2001 statistics provided by the authors in lines 53-55 appear outdated. Authors should provide more recent figures, say 2015, as they did for the subsequent sentence in lines 55-56.

Response:- We used 2001 data-based report and know we corrected it. 

The sentence on line 60 needs to be revised for clarity and correctness.

Response:- Based on the comment, we corrected the statements. 

Materials and Methods

In lines 66-71, while the authors did well by describing the geography of the study setting, it will be beneficial for them to also provide a few demographic descriptions, particularly the estimated population size of the study area.

Response:- Based on the suggestion, we included the population of the HDSS. 

In lines 86-88, it is not clear how the authors used Stata to randomly select participants from each stratum. Authors should be able to describe this process was done.

Response:- Arba Minch HDSS is a surveillance site where all the list of individuals in the site are registered with a unique identification (for both the individuals and house). So, we got this list and developed a sampling frame. Here we had 8 sampling frames (for each age group and sex categories). Using STATA software, we randomly selected the study subjects (individuals). 

Lines 104-105. This sentence should be revised into a better form.

Response:- Based on the suggestion, we corrected 

Authors did not describe all their variables of interest. It is more appropriate for authors to provide a separate sub-section to describe their outcome and predictor variables and their measurements. For convenience, authors should consider labelling BMI as underweight (<18.0kg/m2), normal (18.0–24.9kg/m2), and overweight (≥25kg/m2) instead of the real measurements.

Response:- Under Data collection instruments subsection of the methods, we described the method we followed for the measurement of the variable. And we indicated the methods we follow for generating new variables (based on the collected data) for analysis. Of course, some of the variable were missed and know we make the modification based on your recommendation. 

Authors stated that they used a p-value of 0.25 for the bivariate analysis but used 0.05 for the multivariate analysis. We know 0.25 is 75% while 0.055 is 95% CI. Authors should explain why they used these two different standards.

Response:- We did not decide whether there is association between outcome and predictors based on the p-value less than 0.25. This cut-off point (p-value less than 0.25) is used only to consider the predictor variable as a part for multivariate analysis. To define a variable (for both bivariate and multivariate analysis) as identified predictor for the outcome variable, we consider p-value less than 0.05 for both bivariate and multiple logistic regressions analysis.

Authors have stated that they have assessed multicollinearity using variable inflation factors. Authors should provide the resultant VIF value like they did for the Hosmer and Lemeshow goodness of fit test.

Response:- We corrected by including the maximum VIF value 

Results

The presentation is generally okay but there are a few issues that need to be addressed.

In line 152, authors should rather say “more than one out of ten” instead of saying “nearly one out of ten”.

Response:- Based on the comment, we corrected it. 

In line 155, authors seem to want to say, “fresh food” instead of “flesh food”. Also, there appear to be no fresh food in Table 2 as stated by the authors.

Response:- We were explaining about meet consumption by says ‘flesh’. Now we changed to meet. 

Authors should avoid using prevalence in the multivariate analysis section as can be found in line 155.

Response:- We corrected throughout the document 

In line 182, authors should use “age groups” instead of “groups of age”.

Response:- Based on the comment, we corrected it. 

In line 184, authors should delete the “to” in the “concerning to”.

Response:- Based on the comment, we corrected it. 

Lines 187, 236, 277: Authors should avoid using “probability” and “prevalence” as they worked with only odds ratios.

Response:- We corrected throughout the document 

For easy visibility and convenience, authors should use asterisks (*) to indicate the significant values in Table 3.

Response:- Based on the comment, we corrected it. 

Discussion

In line 203 and 204, revise the sentence. Use “have reported” instead of “were reported”.

Authors should revise the sentence in lines 207 and 208 as there is something wrong with it.

Response:- Based on the comment, we corrected it.

Line 209: Ethiopians instead of Ethiopian.

Response:- Based on the comment, we corrected it.

Lines 210, 214, 230, 231, 233, 239, 254: Avoid prevalence here. Odds, chances, and likelihood are examples author may use.

Response:- We corrected throughout the document 

Line 215: Iranians instead of Iranian.

Response:- Based on the comment, we corrected it.

Line 216, 217-218: Those who never married, not those never married.

Response:- Based on the comment, we corrected it.

Line 218 and others; Authors should explain the possible (speculative) reasons for their findings. For example, why would married women have low hypertension as found in your study? This should be done for the significant variables discussed in this section. Where this is not obvious, authors could briefly explain what could have led to this finding?

Response:- Based on the comment, we corrected it.

Line: 264: Reduced risk instead of reduced the risk.

Response:- Based on the comment, we corrected it.

Lines 266-267: Authors should either state and explain these factors or delete this statement as it appears incomplete and needless.

Response:- Based on the comment, we corrected it.

Lines 268-271: Authors should revise these sentences very well as they appear to be incomplete and unclear.

Response:- Based on the comment, we corrected it.

Conclusion

Lines 273-275: Authors should adequately revise this sentence as it is unclear.

Response:- Based on the comment, we corrected it.

General comments

While the paper has addressed an important public health issue and will contribute to the literature, it is noteworthy that the paper is also laden with some grammatical and diction challenges that have constrained the quality and flow of the paper. These are mainly found in the introduction and the discussion sections. Hence, this paper should be adequately proofread and revised accordingly before it be publishable.

---

## [Editor Report · Decision Letter 1]

16 Jun 2020

PONE-D-20-02687R1

Prevalence of hypertension and associated factors among adult residents in Arba Minch Health and Demographic Surveillance Site, Southern Ethiopia.

PLOS ONE

Dear Mr. Gutema,

Thank you for submitting your manuscript to PLOS ONE. I participated as a reviewer for the initial evaluation of this manuscript and after careful consideration of your revision, we feel that it has merit but does not fully meet PLOS ONE’s publication criteria and some review comments have not been addressed as it currently stands. Therefore, we invite you to submit a revised version of the manuscript that addresses the points raised during the review process.

We look forward to receiving your revised manuscript.

Kind regards,

Samuel H. Nyarko, PhD

Academic Editor

PLOS ONE

**Journal Requirements:**

Authors have effectively addressed the majority of the reviewer and editorial comments. However, there still remain a few important issues to be addressed.

Requirement 1: a) Authors byline and contributorship were not fully provided on the cover page (See cover page format). b) Authors level 2 headings do not abide by the “bold and font” format of the journal. c) The format style of the journal requires that authors provide unstructured abstract.

Requirement 2: Authors have stated that they decided to upload the data set with the manuscript. However, this cannot be found.

Reviewers' comments and evaluation:

**Reviewer 1**

1. Comment 3: Authors did not address this comment as they stated (see Line 57 of track changes) and should be addressed.

**Reviewer 2**

Introduction

2. Little has been done by authors to make the statement: “the leading cause of premature age-standardized mortality rates” (lines 42 and 43 of original version) meaningfully clear. Authors should recast this statement to make it clearer.

3. Authors have addressed the comment of the reviewer (originally line 60); however, the second statement still appears incomplete (Lines 66-68) and should be revised to make it complete. 

Materials and Methods

4. Authors have provided the recommended information regarding lines 66-71, but have to provide citation for such report.

5. Authors provided a cogent explanation to the comment regarding the sampling procedure (Lines 86-88) but authors should incorporate it in the manuscript for readers to clearly understand.

6. Authors did not fully address this comment regarding the description of the variable of interest. Also, the BMI categories such as underweight, normal and overweight appear only in the methods section but did not reflect in the tables and the analysis. These should be addressed by the authors

Discussion

7. The second comment (lines 207-208) has not been addressed as stated by the authors in their response  letter.  Authors are invited to address this. 

General comments

8. Authors did not respond to the comment as to whether they have proofread and revised the manuscript to make it publishable. I invite authors  to address this issue.

---

## [Author Response · Author response to Decision Letter 1]

20 Jun 2020

Journal Requirements:

Authors have effectively addressed the majority of the reviewer and editorial comments. However, there still remain a few important issues to be addressed.

Requirement 1: a) Authors byline and contributorship were not fully provided on the cover page (See cover page format). b) Authors level 2 headings do not abide by the “bold and font” format of the journal. c) The format style of the journal requires that authors provide unstructured abstract.

Response� We corrected according to the Journals requirement 

Requirement 2: Authors have stated that they decided to upload the data set with the manuscript. However, this cannot be found.

Response� We uploaded the minimum dataset. 

Reviewers' comments and evaluation:

Reviewer 1

1. Comment 3: Authors did not address this comment as they stated (see Line 57 of track changes) and should be addressed.

Response: Thank you for the comment. We noted that and corrected it. 

Reviewer 2

Introduction

2. Little has been done by authors to make the statement: “the leading cause of premature age-standardized mortality rates” (lines 42 and 43 of original version) meaningfully clear. Authors should recast this statement to make it clearer.

Response: Thank you for the comments. Yes, we noted that and correct 

3. Authors have addressed the comment of the reviewer (originally line 60); however, the second statement still appears incomplete (Lines 66-68) and should be revised to make it complete. 

Response: Yes, we noted that and correct 

Materials and Methods

4. Authors have provided the recommended information regarding lines 66-71, but have to provide citation for such report.

Response: We get from the database of the Arba Minch HDSS. We got the data while during the sampling frame. But we also included the district’s projected population of 2017, which have also reference. 

5. Authors provided a cogent explanation to the comment regarding the sampling procedure (Lines 86-88) but authors should incorporate it in the manuscript for readers to clearly understand.

Response: Noted and made some correction. 

6. Authors did not fully address this comment regarding the description of the variable of interest. Also, the BMI categories such as underweight, normal and overweight appear only in the methods section but did not reflect in the tables and the analysis. These should be addressed by the authors

Response: Regarding BMI cut-off points, the data were categorized with the cutoff of this one (18.5). The sub-title is incorporated. 

Discussion

7. The second comment (lines 207-208) has not been addressed as stated by the authors in their response letter. Authors are invited to address this. 

Response: - Based on the comment, we corrected it.

General comments

8. Authors did not respond to the comment as to whether they have proofread and revised the manuscript to make it publishable. I invite authors to address this issue.

Response: - Yes, we read the whole documents and corrected identified errors. 

The reviewer gave important comments. We modify based on the comment given. In addition, we attached document with trace changes and the revised.

---

## [Editor Report · Decision Letter 2]

7 Jul 2020

PONE-D-20-02687R2

Prevalence of hypertension and associated factors among adult residents in Arba Minch Health and Demographic Surveillance Site, Southern Ethiopia.

PLOS ONE

Dear Mr. Gutema, 

Thank you for submitting your manuscript to PLOS ONE. I participated as a reviewer for the initial evaluation of this manuscript and after careful consideration of your revision, we feel that it has merit but does not fully meet PLOS ONE’s publication criteria as it currently stands. Therefore, we invite you to submit a revised version of the manuscript that addresses the points raised during the review process. 

We look forward to receiving your revised manuscript.

Kind regards,

Samuel H. Nyarko, Ph.D.

Academic Editor

PLOS ONE

Editor comments:

Authors have effectively addressed virtually all the comments. However, a few important issues remain to be addressed.

Introduction

Sentence in Lines 65-66 appears to be a repetition of that in Lines 50-51. Authors should delete sentence in lines 65-66: In Ethiopia, hypertension was one of the leading causes of age standardized death rates in 2015.

Tables

Authors should replace the figures with the BMI categories such as underweight, normal and overweight as provided in the methods and the results sections to ensure consistency and easy reference.

Discussion

Lines 257: Ethiopia, not Ethiopian.

Lines 287-289 should read: On further analysis, the relationship is consistent with that of some high-income countries (countries from USA/Canada and Europe) but inverse for some African countries.

---

## [Author Response · Author response to Decision Letter 2]

17 Jul 2020

PONE-D-20-02687R1

Prevalence of hypertension and associated factors among adult residents in Arba Minch Health and Demographic Surveillance Site, Southern Ethiopia.

Editor comments:

Authors have effectively addressed virtually all the comments. However, a few important issues remain to be addressed.

Introduction

Sentence in Lines 65-66 appears to be a repetition of that in Lines 50-51. Authors should delete sentence in lines 65-66: In Ethiopia, hypertension was one of the leading causes of age standardized death rates in 2015.

Response� Done according to the recommendation 

Tables

Authors should replace the figures with the BMI categories such as underweight, normal and overweight as provided in the methods and the results sections to ensure consistency and easy reference.

Response� Done according to the recommendation

Discussion

Lines 257: Ethiopia, not Ethiopian.

Response� Done according to the recommendation

Lines 287-289 should read: On further analysis, the relationship is consistent with that of some high-income countries (countries from USA/Canada and Europe) but inverse for some African countries.

Response� Done according to the recommendation 

Thank you for your comment. We modify based on the comment given. In addition, we attached document with trace changes and the revised.

---

## [Editor Report · Decision Letter 3]

24 Jul 2020

Prevalence of hypertension and associated factors among adult residents in Arba Minch Health and Demographic Surveillance Site, Southern Ethiopia.

PONE-D-20-02687R3

Dear Mr. Gutema,

We’re pleased to inform you that your manuscript has been judged scientifically suitable for publication and will be formally accepted for publication once it meets all outstanding technical requirements.

Kind regards,

Samuel H. Nyarko, Ph.D

Guest Editor

PLOS ONE

---

## [Editor Report · Acceptance letter]

30 Jul 2020

PONE-D-20-02687R3 

Prevalence of hypertension and associated factors among adult residents in Arba Minch Health and Demographic Surveillance Site, Southern Ethiopia. 

Dear Dr. Gutema:

I'm pleased to inform you that your manuscript has been deemed suitable for publication in PLOS ONE. Congratulations! Your manuscript is now with our production department. 

Kind regards, 

on behalf of

Dr. Samuel H. Nyarko 

Guest Editor

PLOS ONE